# Fluid-Structure Coupling Creep Characteristics of Red-Bed Soft Rock in South China

**Shuguang Zhang** [1,2,*] **, Xiao Yun** [1] **, Yu Song** [1,*] **, Wenbo Liu** [3] **and Li Yang** [4]

1   School of Civil Engineering, Guilin University of Technology, Guilin 541004, China
2   Guangxi Key Laboratory of Geomechanics and Geotechnical Engineering, Guilin 541004, China
3   State Key Laboratory of Geomechanics and Geotechnical Engineering, Institute of Rock and Soil Mechanics, Chinese Academy of Sciences, Wuhan 430071, China
4   College of Foreign Languages, Guilin University of Technology, Guilin 541004, China
*   Correspondence: zhangsg@glut.edu.cn (S.Z.); songyu119@glut.edu.cn (Y.S.); Tel.: +86-15004184898 (S.Z.); +86-18877335968 (Y.S.)

**Abstract:** In order to study the creep characteristics and mechanism of red-bed soft rock under the water–rock interaction, fluid–structure coupling triaxial compression tests and creep tests under stepwise loading were carried out. Furthermore, the influences of seepage pressure and stress on creep deformation, long-term strength, Poisson's ratio, and seepage velocity were analyzed. According to the experimental results, the influence of seepage on the creep of soft rock cannot be ignored. The results show that the seepage leads to a decrease in triaxial strength and long-term strength, and an increase in instantaneous deformation and creep deformation. The failure mode of triaxial compression changes from shear failure to tension-shear conjugate failure, whereas the long-term strength of fluid–structure coupling creep is 60%~70% of the triaxial strength. When the stress level and seepage pressure are relatively small, the Poisson's ratio of creep increases with the increase of seepage pressure, and the radial creep deformation response lags behind the axial creep deformation. However, at a high stress level and osmotic pressure, the Poisson's ratio and seepage velocity increase rapidly, and the duration of the accelerated creep is obviously shortened. Through the analysis of the influence of seepage pressure on the seepage velocity, with the increase in the seepage velocity, the seepage velocity changes and fluctuations are more obvious, which further confirms the damaging effect of seepage pressure and erosion on the internal structure. In the field monitoring of actual engineering, the rapid change of seepage velocity can be used as a precursor signal to predict the instability. Therefore, the water–rock interaction cannot be ignored in the analysis of mechanical properties and long-term stability of red-bed soft rocks.

**Keywords:** creep characteristics; fluid–structure interaction; creep model; red-bed soft rock; seepage pressure

## 1. Introduction

Red-bed soft rock is a special soil often encountered during engineering construction in South China, and its typical characteristics are obvious water–physical and rheological properties. Under the action of frequent rainfall and groundwater, its microstructure and engineering properties are seriously weakened. When this weakening is coupled with the action of construction disturbance and external load, it is easy to induce engineering geological disasters such as landslides and collapse. Therefore, it is necessary to study the fluid–structure coupling creep characteristics of red-bed soft rocks [1].

Domestic and foreign scholars' research on the water–physical properties of soft rocks mainly focus on the deterioration of mechanical properties, water softening, and disintegration. Studies on the strength characteristics, deformation and failure characteristics, and microstructure changes of water-bearing soft rocks have determined that the main reasons for the softening of soft rocks in the presence of water are as follows: (1) water easily leads

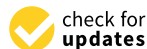



to the dissolution and destruction of argillaceous cement among mineral particles in rock samples and (2) the expansion and decomposition of clay minerals by water absorption, which makes the rock structure more porous and loose and the pores mostly filled with water [2,3]. The disintegration mechanism of soft rock is due to the hydration, diffusion, and loss of clay particles on the water–rock interface in the argillaceous filling area under the action of water, which leads to a reduction in the argillaceous cementation zone and a decrease in the cohesion of rock and soil mass. Therefore, the sensitivity to water is an important factor to be considered in the study of soft rock stability [4,5].

The rheological properties of soft rocks are mainly based on experimental research and rheological models. Laboratory rheological tests are the main means to obtain the rheological properties of rocks, which have gradually developed from uniaxial compression creep experiments to triaxial compression creep experiments, shear creep experiments, and water–rock coupled creep experiments, etc. [6–11]. Considering the structural plane and size effect of engineering rock mass, in situ rheological tests are also carried out [12,13]. Because of the large manpower and material input, high technical difficulty, and long test period, there are few experimental studies in this field. In contrast, research on rock rheological constitutive models is fruitful, and the nonlinear viscoelastic–plastic rheological constitutive model of rock is one of the hot topics. In most of these creep models, a variety of nonlinear rheological constitutive equations were established by improving the Nishihara model and the Burgers model, such as the creep damage model, nonlinear viscoelastoplastic rheological model, and accelerated creep model [14–17]. Furthermore, many scholars apply the theory of fractional calculus to the rheological constitutive model of rocks. As the fractional time derivative is a differential-integral convolution operator, which fully reflects the historical dependence of the development of system functions, it can be used to describe the creep deformation law of rock and soil mass [18–20]. All the above studies have laid a good foundation for exploring the rheological properties of red-bed soft rocks.

In addition, the water–rock interaction and its influence on the rheological properties of soft rocks cannot be ignored. In the process of the water–rock interaction, the repeated rise and fall of water pressure and the dry–wet cycle cause irreversible cumulative damage to the rock mass. The red-bed soft rock transforms from a relatively dense structure into a loose porous structure with micro-cracks and pores, which is shown in the deterioration of the compressive and shear strengths and a change in the failure mode at the macro-level [21]. At the same time, under the combined action of osmotic water pressure and external force, creep deformation further aggravates the instability and failure of red-bed soft rock and then produces engineering geological disasters such as collapse, landslide, and debris flow [22]. Xie et al. studied the mechanical properties of soft rocks under the action of water and rock, compared and analyzed the microstructure of soft rocks under different water cut states, and discussed the softening mechanism of red-bed soft rocks underwater [23]. Yu et al. obtained the theoretical seepage threshold by constructing the renormalization group model of soft rocks and established the seepage damage constitutive model of red-bed soft rocks [24]. Using the MMF model, Zhang and Huang et al. derived the disintegration evolution model of red-bed soft rock collapse, put forward a new calculation formula for the disintegration ratio, and analyzed the influence of the dry–wet cycle on disintegration and fragmentation [25,26]. Li et al. studied the shear–creep characteristics of red-bed soft rocks in an acidic environment and established an improved Burgers model [27]. Liu et al. conducted saturation tests and numerical simulations of red sandstone under rapid and turbulence modes, discussed the chemical, physical, and mechanical effects of different flow modes on softening, and revealed the influencing mechanism of different flow modes on the softening of red sandstone [28]. Zhou et al. considered the influence of the long-term water–rock interaction on the mesoscopic structure evolution law, introduced pore structure parameters, and established a porosity evolution model of red-bed soft rock [29]. Thus, the influence of water on the rheological properties of red-bed soft rocks has attracted the attention of many scholars, and some research has been carried out on rheological models

and the water–rock interaction and mechanism. However, there are few studies on the convection–solid coupling creep properties.

In this paper, triaxial creep experiments on red-bed soft rocks under different seepage pressures are carried out. Based on the isochronous stress–strain curve and creep rate time–change curve, the long-term strength of red-bed soft rocks is analyzed. The creep Poisson's ratio curve of red-bed soft rocks under different seepage pressures is obtained, and the influence of seepage pressures on the creep deformation characteristics of the rock samples is analyzed. The purpose of this paper is to explore the fluid–structure coupling creep characteristics of red-bed soft rocks, which helps to reveal the coupling mechanism of seepage and internal structure and provides a reference for practical engineering stability analysis and geological disaster prediction.

## 2. Physical and Mechanical Properties of the Samples

### 2.1. Physical Property

Red-bed soft rocks are sediments formed from the continental basins from the Cretaceous to the Tertiary periods in South China. The samples are dark red because they contain a certain amount of pyrite, and they are mostly conglomerates, mudstone, shale, sandstone, and so on. Because the uniaxial strength of saturated rock is usually lower than 30 MPa, these sediments are called red-bed soft rocks. The collected samples were cut into cylinders with a diameter of 50 mm and a height of 100 mm, and the samples with obvious appearance defects are removed. Then, the surface was polished smooth to make standard samples, and the ultrasonic test and weighing test were carried out. The samples with similar wave velocities and weights were selected to eliminate differences as much as possible (Figure 1).

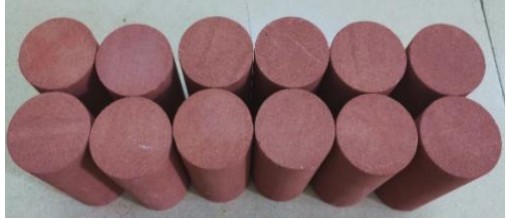

**Figure 1.** Some red-bed soft rock specimens.

The samples are mainly composed of clastic minerals and clay minerals, including quartz, albite, calcium carbonate, montmorillonite, and illite. The XRD test was used to obtain the diffraction pattern and content of each component as shown in Figure 2. The dry density and porosity of the samples were 2.19–2.26 g/cm$^3$ and 6.58–7.01%, respectively.

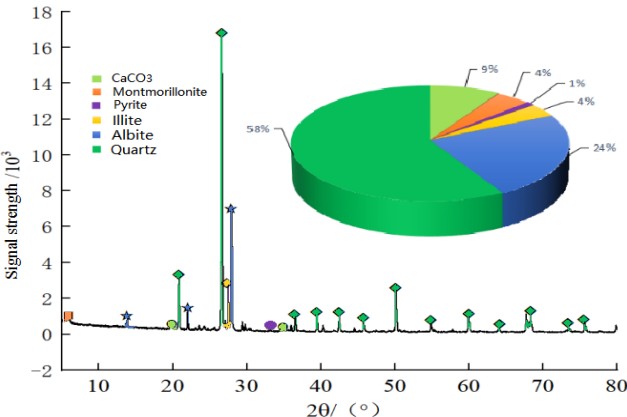

**Figure 2.** XRD pattern and mineral composition.

### 2.2. Strength Properties

The strength of the samples is tested using the automatic rock triaxial test system with a maximum axial force and radial pressure of 2000 kN and 60 MPa, respectively (Figure 3). The equipment has two sets of dynamic and static loading systems, which can carry out rock mechanics tests such as conventional uniaxial and triaxial compression tests, permeability tests, and creep tests of rock materials.

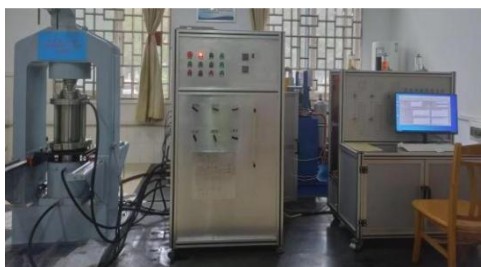

**Figure 3.** The automatic rock triaxial test system.

Before the experiment, the samples were placed in distilled water and forced to saturate using a vacuum saturator. The confining pressure is set at 3 MPa, and the seepage pressures are set at 0 MPa, 0.5 MPa, 1 MPa, and 1.5 MPa. At the beginning of the test, the confining pressure was applied at a loading rate of 0.05 MPa/s, and then the axial load was applied at a loading rate of 0.5 MPa/s. According to the test rules of the International Society of Rock Mechanics, the triaxial compression tests under different seepage pressures were carried out, and the stress–strain curves are shown in Figure 4.

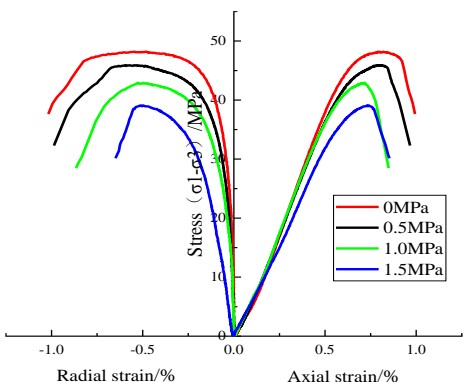

**Figure 4.** Triaxial stress–strain curves under different seepage pressures.

Under different seepage pressures, the triaxial stress–strain curves of the red-bed soft rock show the stages of compaction, elasticity, plastic yield, and strain softening. In the ordinary triaxial test, there is only one obvious penetrating fracture surface, and the failure mode is a compressive shear failure. When the seepage pressure is applied, the elastic modulus, strength, and maximum strain decrease with the increase in the seepage pressure, whereas the number of main cracks increases. The failure mode changes to tensile and shear conjugate failure (Figure 5).

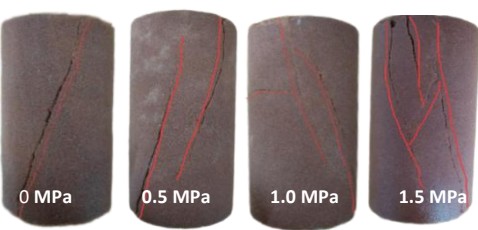

**Figure 5.** Failure modes under different seepage pressures.

### 3. Creep Experiment and Analysis of Test Results

*3.1. Creep Strain–Time Curve*

The creep test uses the stepwise loading method, and the main points of the experiment are as follows:

(1)    The confining pressure remained unchanged at 3 MPa, whereas the seepage pressure was set at 0.5 MPa, 1 MPa, and 1.5 MPa. The liquid used for the seepage is distilled water.

(2)    At the loading rate of 0.05 MPa/s, the axial pressure and confining pressure are applied to the predetermined value, and then the axial load is loaded to the predetermined value.

(3)    The initial load is 50% of the strength of the rock, then it increases by 10% for each of the grade loads.

(4)    For the last grade loads, 80% of the triaxial strength is applied when there is no seepage, and 75% of the triaxial strength is applied when there is seepage.

(5)    The duration of each stage of the load is 24 h and so on until the creep failure occurs.

(6)    The test data were converted into creep strain–time curves using the Bozltmen superposition method (Figure 6).

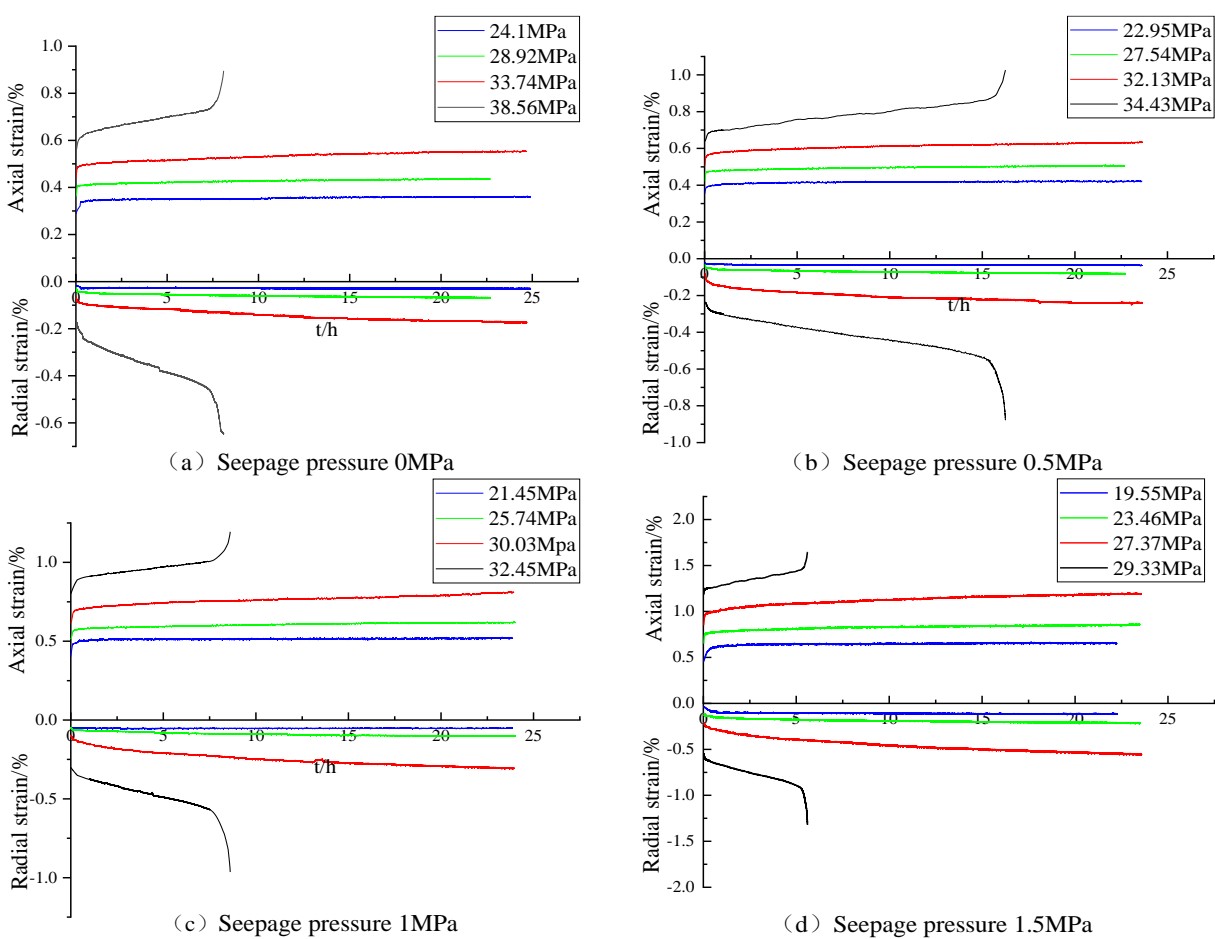

**Figure 6.** The strain–time creep curves.

The creep deformation characteristics of red-bed soft rocks are roughly similar, but the instantaneous deformation and creep deformation increase with an increase in seepage pressure. Under the same seepage pressure, the instantaneous deformation is greater than the creep deformation when the stress is low, whereas the opposite is true when the stress is high. The main reason is that the higher the seepage pressure, the stronger the erosion

effect of water, which intensifies the softening of particles and the enlargement of pores, leading to a larger strain.

When the applied stress level is 70% of the triaxial strength, the creep deformation under different seepage pressures is shown in Table 1. With the increase in seepage pressure, the accelerated creep stage is gradually shortened, but the creep deformation is increased. Taking the seepage pressure of 1.5 Mpa as an example, shortly after the last level of the load is applied, the sample enters the accelerated creep stage, and the creep failure occurs after the creep lasts only about 5 h. The results show that seepage pressure and scour have obvious effects on creep characteristics.

**Table 1.** Creep strain under different seepage pressures.

| Seepage Pressures | 0 MPa | 0.5 MPa | 1.0 MPa | 1.5 MPa |
|---|---|---|---|---|
| Axial deformation/% | 0.123 | 0.134 | 0.192 | 0.347 |
| Axial instantaneous deformation/% | 0.038 | 0.043 | 0.052 | 0.081 |
| Axial creep deformation/% | 0.085 | 0.091 | 0.140 | 0.266 |
| Radial deformation/% | −0.108 | −0.156 | −0.203 | −0.204 |
| Radial instantaneous deformation/% | −0.009 | −0.020 | −0.016 | −0.004 |
| Radial creep deformation/% | −0.099 | −0.137 | −0.188 | −0.201 |

### 3.2. Creep Rate–Time Curve

By sorting out the axial creep data of samples, the creep rate–time curve and the whole process creep curve under different seepage pressures are obtained (Figure 7).

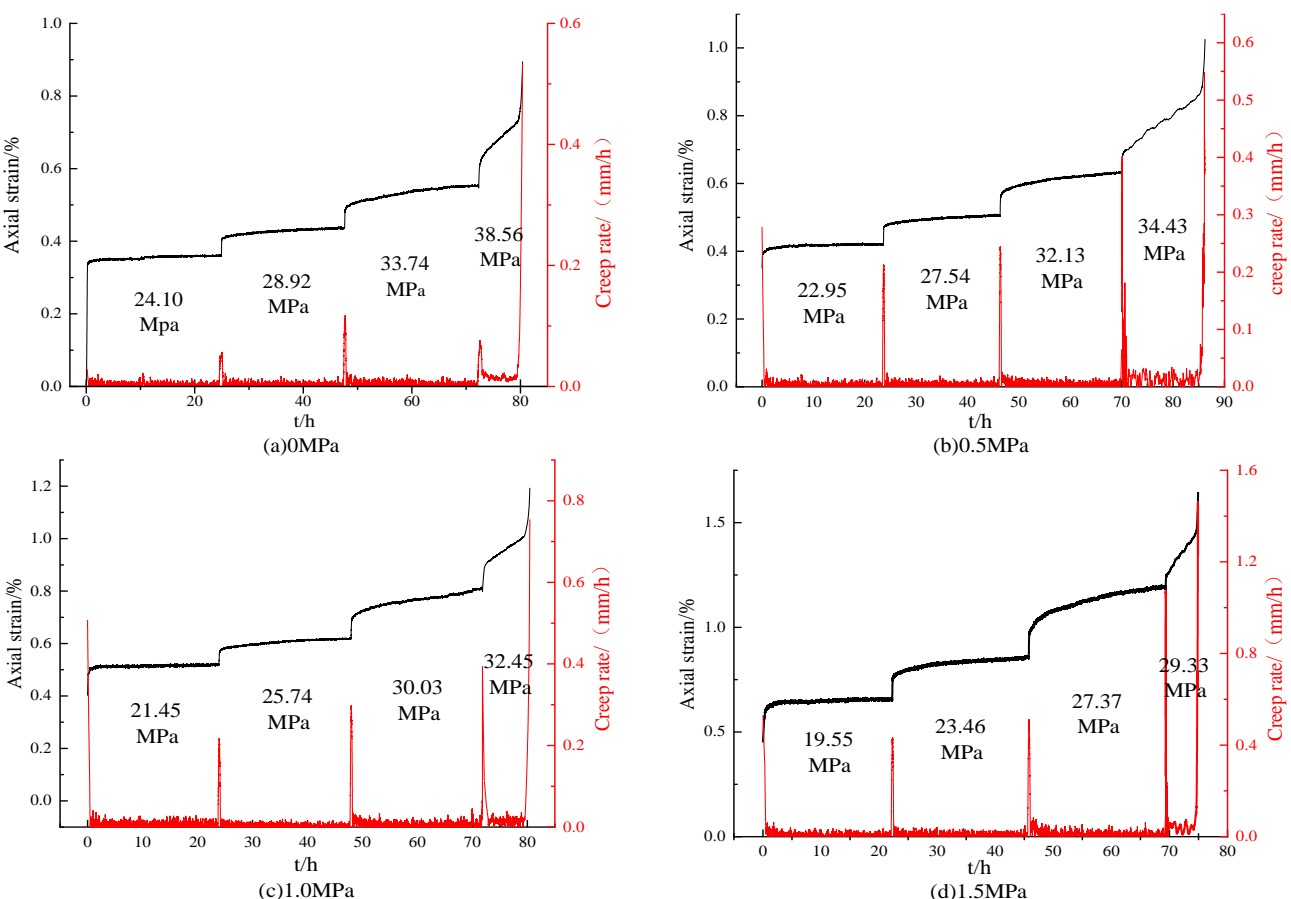

**Figure 7.** Creep rate–time curve under different seepage pressure.

Based on Figure 7, the main results are as follows:

1. Without seepage, the creep curve is gentle, and the creep velocity is stable under the first- and second-grade loads. Under the third-grade loads, the creep deformation obviously increases, whereas the creep velocity decreases gradually until it drops below 0.0015 mm/h, which indicates that the creep tends to be stable. At the fourth-grade loads, the creep rate increases significantly, and the creep strain increases rapidly, eventually leading to creep failure.

2. When the seepage pressure is 0.5 MPa, the creep velocity curve is similar to that without seepage. The creep strain and rate are greater than those without seepage, and the samples were broken after about 16 h.

3. When the seepage pressure is 1.0 MPa, the creep deformation of the first- and second-stage loads increases slightly, the curve is relatively gentle, and the creep velocity finally drops to 0.00027 mm/h and 0.00078 mm/h, respectively. After the third-stage loads are applied, the creep curve shows an obvious upward trend, and the final creep rate is 0.0051 mm/h. Failure occurs when the fourth-grade loads last for about 9 h.

4. When the seepage pressure is 1.5 MPa, the creep velocity curve is similar to that of 1.0 MPa. The creep rate of the first-, second-, and third-stage loads eventually drop to 0.0007 mm/h, 0.00069 mm/h and 0.0067 mm/h, respectively. Failure occurs when the fourth-grade loads lasts for about 5 h.

### 3.3. Long-Term Strength

By sorting out the axial creep data of the rock samples, the isochronal stress–strain curves of red-bed soft rock under different seepage pressures are obtained (Figure 8).

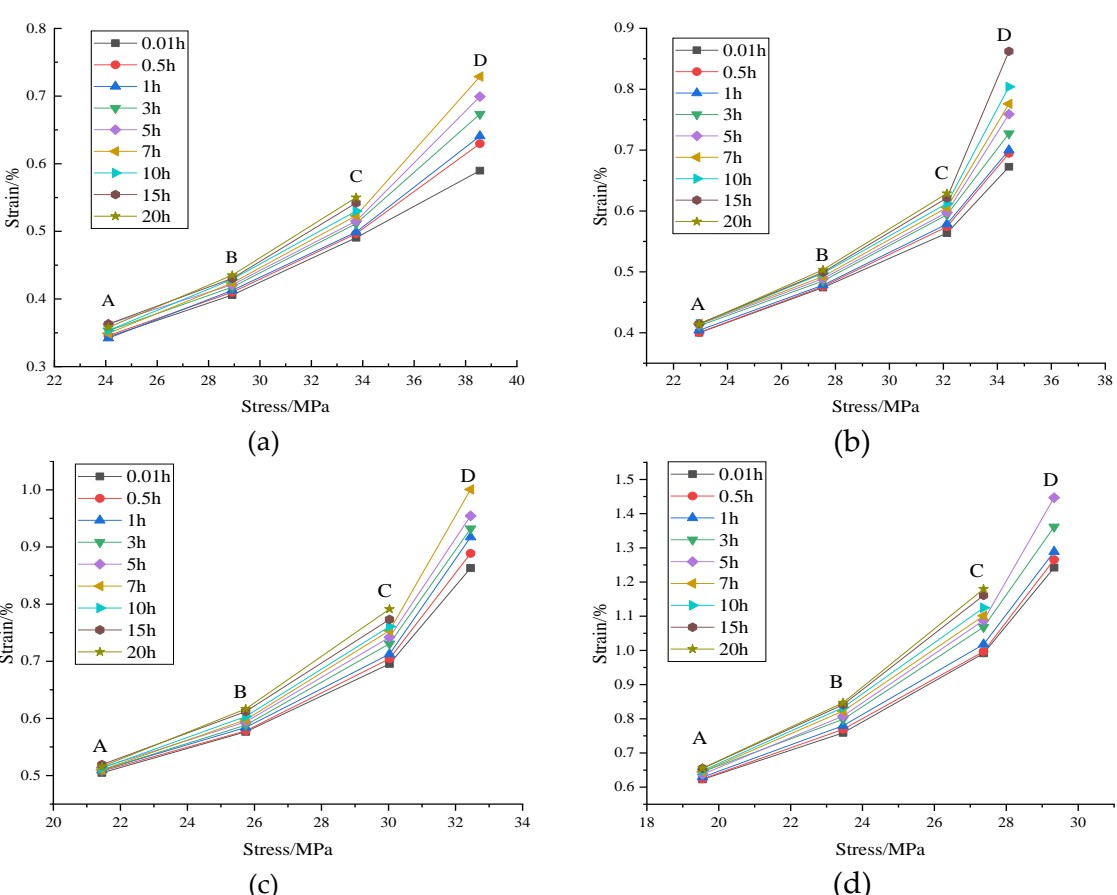

**Figure 8.** Isochronous stress–strain curves under different seepage pressures. (**a**) Seepage pressure of 0 MPa. (**b**) Seepage pressure of 0.5 MPa. (**c**) Seepage pressure of 1.0 MPa. (**d**) Seepage pressure of 1.5 MPa.

When the seepage pressure is less than 0.5 MPa, the isotropic stress–strain curves of the first- and second-grade loads are almost stable at points A and B (the variation range is less than 0.03%), and the strain is mainly caused by elastic deformation. When the third-grade loads are applied (($\sigma_1 - \sigma_3$) = 33.74 MPa), the strain begins to diverge obviously at point C. As time increases, the strain curve gradually becomes sparse, and the strain increase amplitude becomes larger. When the fourth-grade loads are applied (($\sigma_1 - \sigma_3$) = 38.56 MPa), the accelerated creep phase gradually enters at point D, and the strain increases rapidly and eventually fails. Therefore, the long-term strength can be considered to be between the third- and fourth-grade loads, corresponding to about 70% of the maximum deviator stress.

When the seepage pressure is 1.0 MPa and 1.5 MPa, the isotropic stress–strain curves have similar variation rules. The difference is that obvious strain divergence appears at point B. When the third-grade loads are applied, the strain changes with time and the creep rate cannot be stabilized, and accelerated creep failure occurs shortly after at point D. Therefore, it can be considered that the long-term strength value is lower than the third-grade loads and is determined to be about 60% of the maximum deviatoric stress.

Based on the creep rate–time curves, the long-term strength under different seepage pressure conditions is shown in Table 2.

**Table 2.** The long-term strength under different seepage pressure.

| Seepage Pressures | 0 MPa | 0.5 MPa | 1.0 MPa | 1.5 MPa |
|---|---|---|---|---|
| Maximum deviator stress/Mpa | 48.2 | 45.9 | 42.9 | 39.1 |
| Long-term strength/Mpa | 33.74 | 32.13 | 25.74 | 23.46 |

### 3.4. Creep Poisson's Ratio Curve

By sorting out the creep data of the rock samples, the creep Poisson's ratio change curve of red-bed soft rock under different seepage pressures was obtained (Figure 9).\

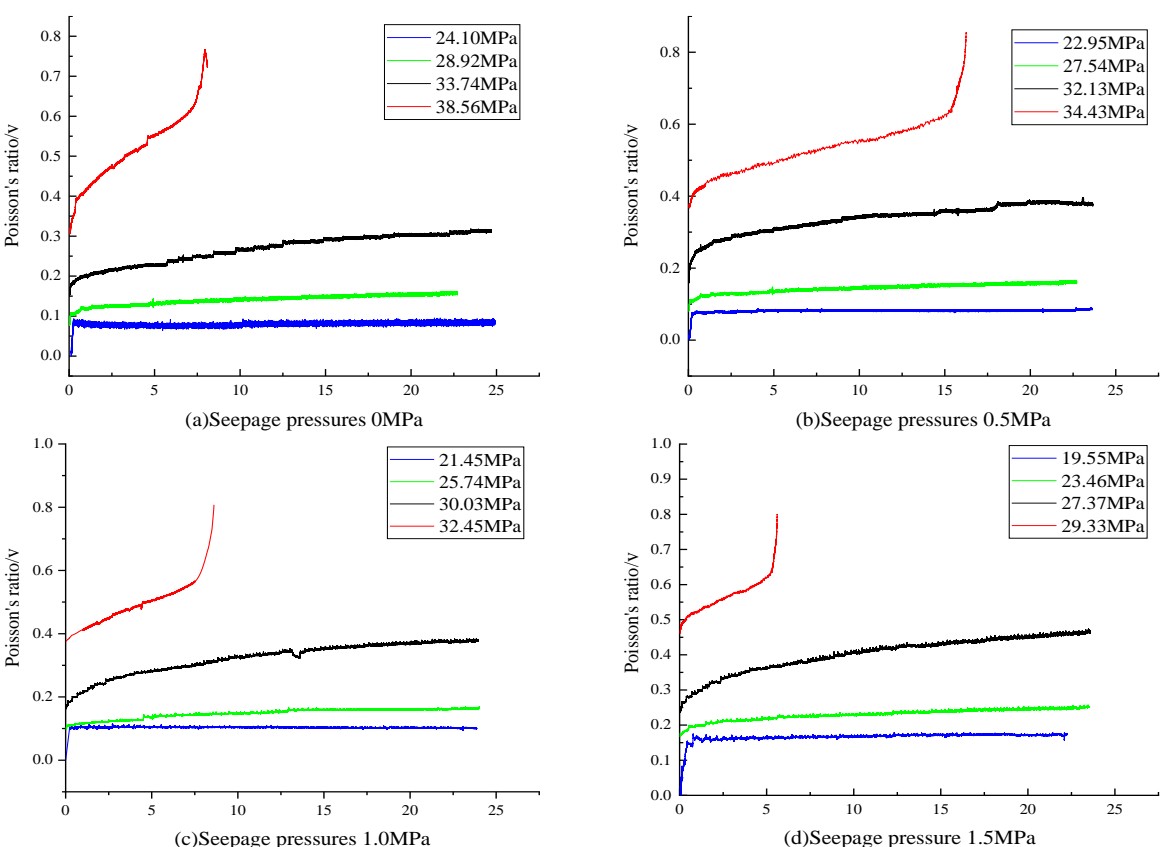

**Figure 9.** Poisson's ratio–time curve under different seepage pressure.

Under the conditions of the first- and second-grade loads, Poisson's ratio will eventually become constant as creep tends to be stable. When the load increases to the third-grade loads, Poisson's ratio under different seepage pressures increases gradually but tends to a relatively stable value, which increases with the increase in seepage pressure, usually between 0.3 and 0.5. This indicates that the response of radial strain lags behind the process of axial strain, and the seepage has an obvious effect on the creep failure of red-bed soft rock. Poisson's ratio increases rapidly when the last level of load is applied, and Poisson's ratio changes faster with the increase in seepage pressure. This shows that the load is the direct cause of the rapid development of radial deformation, and the damage caused by seepage pressure to the internal structure of red-bed soft rock cannot be ignored. These trends must be taken into account in practical engineering.

## 4. Analysis of Seepage Characteristics in the Creep Process

Scholars have carried out a lot of research on the permeability characteristics of rocks in the stress–strain process, including the influence of confining pressure, the influence of the strain and failure mode on permeability, the permeability change and seepage mechanism in the deformation process, etc. According to their research results, the permeability evolution law under triaxial loading corresponds to the stress–strain relationship and has experienced three stages: gradual decrease, steady development, and rapid increase [30–32]. The fluid–rock interaction will cause hydrochemical reactions and erosion disturbances, which will lead to changes in the porosity, permeability, and mechanical stability of rocks [33,34]. Therefore, seepage objectively reflects the change of the internal structure of rock mass under the action of fluid–structure coupling. However, there is limited research on seepage characteristics and the evolution law of permeability in the process of creep.

In this paper, the change of seepage rate with time is obtained by automatically collecting data once per second, and then the evolution law of permeability in the process of fluid–structure coupled creep is studied.

### 4.1. Instantaneous Deformation Stage

For the initial loading stage, the changing relationship between seepage velocity and time is shown in Figure 10a. The seepage characteristics of red-bed soft rock are compared and analyzed according to the seepage test results of Wang et al. in the creep process of hard rock (Figure 10b) [35].

At the initial stage of loading, the rock enters the compaction stage, and the internal pores gradually close with the increase of axial stress. Because the permeability decreases gradually, the seepage velocity also decreases gradually with the increase in time. Although the first level load applied is 50% of the strength, the seepage velocity of red-bed soft rock tends to be steady in about 3 min, whereas that of hard rock is about 15 min, which means that the compaction process of red-bed soft rock is faster. From the process of seepage velocity variation, the variation range of red-bed soft rock is relatively small and has the characteristics of oscillating decline. In addition, the higher the seepage velocity, the longer the stabilization time. This shows that during the loading process, there is not only pore compaction but also dislocation of sheet structure in the red-bed soft rock, which makes the seepage velocity fluctuate to a certain extent, which is different from that of hard rock.

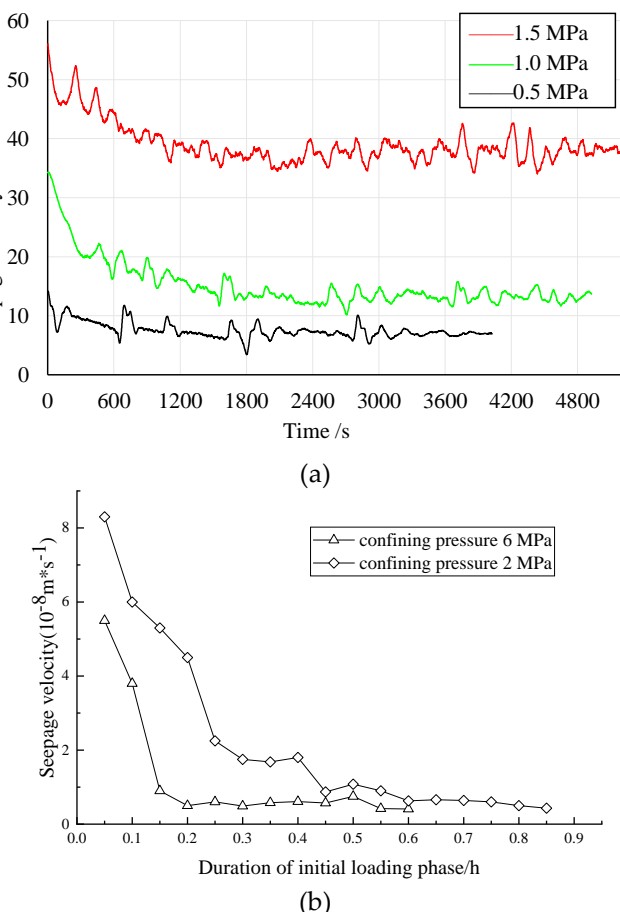

(a)

(b)

**Figure 10.** The seepage velocity–time curve in the initial loading stage. (**a**) Red-bed soft rock. (**b**) Hard rock [35].

### 4.2. Steady Creep and Accelerated Creep Stages

For the steady-state creep and accelerated creep stages, the relationship of rock seepage velocity with time under different seepage pressures is shown in Figure 11.

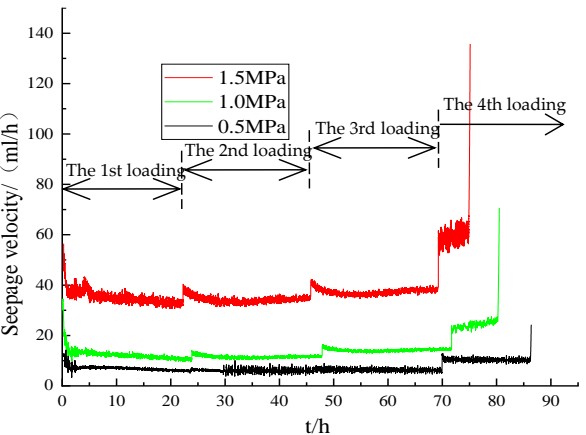

**Figure 11.** Seepage velocity–time curve in the whole process.

After the adjustment of instantaneous deformation, the seepage velocity slowly decreases until it reaches a certain stable value, indicating that the internal structure of the rock reaches a new stable state after compaction. At the instantaneous loading stage of each load, the seepage velocity changes greatly, but at the steady creep stage, the seepage velocity changes little. However, the fluctuation of seepage velocity is more obvious with

the increase of seepage pressure. The main reason for the above phenomenon is that micro-cracks are generated in the rock during the instantaneous loading process. When the rock enters the steady creep stage, the pores and cracks in the sample gradually close again. Meanwhile, the seepage pressure and erosion react to the above process, which leads to the fluctuation of the seepage velocity curve.

Under the fourth-grade loads, the creep curve of rock presents three stages. The change in seepage velocity is relatively stable in the decelerating creep and the stable creep stages. As the applied stress exceeds the long-term strength, it gradually enters accelerated creep with time. At this time, the internal cracks are constantly generated and expanded, and finally connected with each other to form macroscopic cracks, resulting in creep failure of the sample. In this process, the seepage flow gradually changes from pore flow to fracture flow, which leads to an obvious increase in seepage velocity.

### 4.3. Influence of Seepage Pressure on Velocity

In red-bed soft rock, the influence of seepage pressure on seepage velocity is not only reflected in the driving force of the fluid but also in the water–rock interaction, which is specifically shown as follows:

1. When the seepage pressure is 0.5 MPa, the seepage velocity decreases rapidly in the first two hours and then slowly at the first-grade loads. This indicates that the pores of the sample are compressed first, and then the porosity decreases slowly. Under the second- and third-grade loads, the seepage velocity is basically constant after some adjustment, indicating no significant change in porosity.
2. When the seepage pressure is 1.0 MPa, for the first- and second-grade loads, the characteristics of the curve have no obvious change compared with before, only the seepage velocity increases slightly, which is probably caused by the increase of seepage pressure. After applying the third-grade loads, the seepage velocity increases slowly, but not obviously.
3. When the osmotic pressure is 1.5 MPa, the change characteristics of the curve are similar to that of 1.0 MPa, but the seepage velocity increases greatly. This shows that seepage pressure plays a key role in the process of destroying the internal structure of the red-bed soft rock.
4. At the fourth stage of loading, the effect of seepage pressure is more obvious, which is mainly reflected in that the slope of the seepage velocity curve increases with the increase of seepage pressure, and the fluctuation of the curve is intensified.

The essence of the increase of seepage velocity is the increase in porosity, which is consistent with the changes in creep rate, long-term strength, and Poisson's ratio, and is an objective reflection of the changes in the macroscopic index and internal structure of the specimen in the creep process.

## 5. Conclusions

The fluid–structure coupled creep test of red-bed soft rock was carried out, and then the influence of seepage pressure on creep characteristics was analyzed. The main conclusions were drawn as follows:

(1) With the increase of seepage pressure, the failure mode of triaxial compression changes from shear failure to tension–shear conjugate failure. The creep strain increases significantly, and the long-term strength is 60%~70% of the triaxial strength, indicating that seepage pressure and scour have obvious effects on the creep characteristics.
(2) When the stress level and seepage pressure are relatively low, the radial strain response lags behind the axial strain history. However, in the accelerated creep stage, the radial strain is more sensitive than the axial strain, and the duration of creep decreases with the increase of seepage pressure.
(3) The seepage velocity increases significantly with the increase in seepage pressure, indicating that the seepage leads to an increase in porosity and the failure of the internal structure. Especially when the seepage pressure is 1.5 MPa, the seepage

velocity increases greatly. Therefore, the influence of the fluid–structure interaction should be fully considered in the long-term stability analysis of engineering.

(4) In the accelerated creep stage, the slope and fluctuation amplitude of the seepage velocity curve increase significantly with the increase in seepage pressure, which can be used as the precursor signal for monitoring the creep instability in the field.

Water softening and strong creep are typical characteristics of red-bed soft rock. The study of fluid–solid coupling creep characteristics is of theoretical significance for the long-term stability analysis of soft rock engineering and practical significance for the prediction of engineering geological disasters. In this paper, the influence of seepage pressure on creep characteristics is explored through the fluid–solid coupled creep experiment, and the change in the internal structure is analyzed using the seepage velocity in the creep process. The research results also show that the influence of seepage on the creep of soft rock and hard rock is different. Due to the differences of experimental samples, experimental equipment, and the number of tests, the research on the mechanism of fluid–structure interaction of red-bed soft rock is still lacking a more powerful quantitative basis. In particular, the cross-comparative analysis of influencing factors through the construction of fluid–structure coupling creep model needs to be improved.

**Author Contributions:** Project administration, S.Z.; writing—original draft preparation, X.Y.; investigation, Y.S.; visualization, W.L.; writing—review and editing, L.Y. All authors have read and agreed to the published version of the manuscript.

**Funding:** This research was funded by the National Natural Science Foundation of China (Grant No. 42067041).

**Institutional Review Board Statement:** Not applicable.

**Informed Consent Statement:** Not applicable.

**Data Availability Statement:** Not applicable.

**Conflicts of Interest:** The authors declare no conflict of interest.

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
