# Peer review of "Fluid-Structure Coupling Creep Characteristics of Red-Bed Soft Rock in South China"

_water, doi:10.3390/w14244088_

Round 1

Author Response

According to your review comments, we have written the reply document for your review. Thank you very much.

Reviewer 2 Report

(1) How many samples does the author use for each seepage pressure experiment? If it is one, there will be contingency?

(2) In the introduction, the author discussed the current research situation, but did not indicate the importance of this study, there have been many similar studies in this area. What is the innovation of the author's research?

(3) Fluid-structure coupling creep model is proposed in the paper, authors should compare with existing models, or use published data for further verification.

(4) the paper analysis should try to draw quantitative conclusions.

(5)  language needs to be improved

Author Response

(The authors gave the same response as above.)

Reviewer 3 Report

This study should be reconsidered after some revision.

How to get the beautiful fitting curve in Fig. 13?

Please remove Eq. (6) and Section 4.2, and add a discussion section to address the relavent engineering and lab study, and conduct a comparation study and get some advanced understanding about this topic. such as the reference, Chinese Journal of Rock Mechanics and Engineering, 2022, 41(6): 1193-1207. DOI: 10.13722/j.cnki.jrme.2021.0941

Pay attention to writing rules.

Rewrite the Introduction section, make it more scientific and logical.

Author Response

(The authors gave the same response as above.)

Round 2

Reviewer 1 Report

The authors responded point to point my comments. It is with pleasure that i inform you that i do not have any further comments.

Reviewer 2 Report

the revision can be accepted

Reviewer 3 Report

This study can be accepted for publication in this journal.